# Same Incision for Simultaneous Laparoscopic Hand-Assisted Native Nephrectomy Contralateral to the Site of the Kidney Transplant

**DOI:** 10.3390/reports6020029

**Published:** 2023-06-16

**Authors:** Roberta Angelico, Laura Keçi, Laura Tariciotti, Alessandro Anselmo, Evaldo Favi, Tommaso Maria Manzia, Giuseppe Tisone, Roberto Cacciola

**Affiliations:** 1Hepatobiliopancreatic Surgery and Transplant Unit, Department of Surgery Sciences, University of Rome Tor Vergata, 00133 Rome, Italy; roberta.angelico@gmail.com (R.A.); laurakeci@yahoo.com (L.K.); laura.tariciotti@gmail.com (L.T.); alessandro.anselmo@ptvonline.it (A.A.); manzia@med.uniroma2.it (T.M.M.); tisone@uniroma2.it (G.T.); 2General Surgery and Kidney Transplantation, Fondazione IRCCS Ca’ Grande Ospedale Maggiore Policlinico, 20019 Milan, Italy; evaldofavi@gmail.com; 3King Salman Armed Forces Hospital, Tabuk 47512, Saudi Arabia

**Keywords:** kidney transplantation, surgical techniques, native nephrectomy, laparoscopy, mini-invasive surgery, outcomes

## Abstract

Native nephrectomies in patients scheduled for a kidney transplant may represent a major challenge. The timing of the procedures as well as the magnitude of both surgical procedures require a risk mitigation strategy that may be restricted by the specific condition of the patients. We report a case of a simultaneous laparoscopic hand-assisted native nephrectomy contralateral to the site of the living donor kidney transplant.

## 1. Introduction

Native nephrectomy (NN) in patients with end-stage kidney disease (ESKD) undergoing kidney transplantation (KT) has specific indications such as urinary infections, severe hydronephrosis or vesical-ureteral reflux, proteinuria, intractable hypertension, polycystic kidneys or malignancy [1,2,3]. The NN may be performed as a staged procedure (nephrectomy prior to transplantation), simultaneously to the transplant or after the KT [4]. NN as a staged surgery may be performed with a minimally invasive approach, with a small incision; however, it would expose the patient to the risk of suffering from chronic renal insufficiency requiring dialysis due to possible post-operative morbidity before transplantation. When performed simultaneously, the transplant procedure usually requires a larger incision and prolonged operative times, which could expose the patient to a greater risk of graft loss [5]. However, several studies demonstrated that simultaneous NN and KT have excellent outcomes. This has the advantage of only exposing the patient to a single surgical procedure [6]. Moreover, it was recently discovered that this simultaneous approach also provides potential cost savings with no adverse events [7]. NN may be required after KT, but the surgery might compromise the graft function.

The complexity and magnitude of the NN and KT procedures lead doctors to consider risk mitigation strategies [8]. The strategy used in the surgery may be influenced by numerous factors ranging from the indications, the entity of remanent diuresis, the site of the native kidney to be removed, the viability of living related donors and the risks related to both the NN and KT [9].

In this case report, we present the surgical technique of a simultaneous procedure for minimally invasive NN and KT from the same incision, contralateral to the site of the nephrectomy.

## 2. Case Report

A 29-year-old female (weight, 48 kg; body mass index, 18) presented herself to our Transplant Unit as a potential recipient of a living donor KT (LDKT). The patient was mono-nephric due to congenital agenesis of the right kidney. At 18 years of age, the kidney function progressively worsened due to vesical-ureteral reflux in the left kidney, which was associated with hydronephrosis with recurrent pyelonephritis and urosepsis. This condition required numerous surgical, endoscopic and percutaneous procedures of the left kidney, which were performed by urologists of several hospitals across three different countries. Specifically, at the age of 20, an open ureteral reimplantation was performed, but the surgical procedure was followed by a persistence of vesical reflux and urinary infections. To reduce the severity of the reflux, hyaluronic acid/dextranome injections, intermittent catheterization and ureteral stent placement were attempted. However, after the many fails of these endoscopic treatments, a left kidney nephrostomy was placed. Despite multiple procedures, at 28 years of age, the patient developed ESKD and started hemodialysis (HD) via an autologous arteriovenous fistula.

At the recipient’s assessment for KT, the patient presented a urinary infection sustained by multi-resistant *Escherichia coli* bacteria and with no major contraindication for transplantation. At this stage, her husband was presented as a potential living donor of a kidney, and he was put under evaluation for the donation process. This potential donor was considered fit to donate a kidney following the results of the routine evaluation process.

At the multidisciplinary meeting, we determined the surgical strategy to be followed for this specific procedure. In the recipient patient, the nephrectomy of the left native kidney was indicated to treat the potential recurrent urosepsis sustained by persistent *E. coli* infection and severe hydronephrosis. Moreover, the right iliac fossa (RIF) was considered the preferred site for performing the KT due to the substantial risk of finding extensive fibrosis in the left iliac fossa (LIF) and in the retro-peritoneal space due to the previous procedures the patient underwent. Considering the potential benefits of performing a simultaneous NN and KT, we planned a left laparoscopic hand-assisted native nephrectomy (LHA-NN) with a trans-peritoneal approach, placing the hand port in the RIF, and using the same incision to perform the KT.

Due to the chronic urinary infection that was previously mentioned, the patient received a prophylactic course of antibiotics one week before the surgical procedure, obtaining two consecutive negative results.

## 3. Methods and Procedure

The patient was placed in a right lateral supine position and was secured to the operating table. A curved incision with a diameter of 8 cm in the RIF was created to allow a trans-peritoneal approach to the left kidney for NN, and subsequently, a retroperitoneal approach to the right iliac vascular axis for the transplantation.

After opening the peritoneum, the hand port was placed. Then, one camera port (10 mm) and one working port (11 mm) were placed in the LIF and left upper quadrant, respectively (Figure 1 and Figure 2).

Due to previous recurrent pyelonephritis and multiple procedures, we found extensive fibrotic processes surrounding the left kidney and extending distally to the LIF. The left kidney was mobilized from the adhesions. The left ureter was isolated and sectioned between clips. The left renal artery and renal vein were fully dissected and divided using an Echelon Endovascular Stapler Ethicon 30 mm.

The left kidney was extracted through the hand port in the RIF (Figure 3). Then, the peritoneum was closed using a continuous suture with Vicryl 3/0 (Figure 4).

Afterward, the patient was placed in the supine position, and through the same incision via a retroperitoneal approach, the external iliac vessels were prepared for renal transplantation (Figure 5). After preparation of the retroperitoneal fossa, the external iliac artery and vein were exposed, and the lymphatic vessels were ligated.

The left nephrectomy of the donor patient was initiated in the parallel operating room as soon as the NN was completed.

In the recipient patient, the renal graft was anastomosed to the external iliac vessels via terminal-lateral anastomosis with Prolene 5/0. The kidney had an excellent reperfusion and immediate production of urine (Figure 6). The ureter cystostomy was performed using the Lich–Gregoire technique with a double-J ureteral stent insertion [10]. One singular peri-renal drain was positioned. The operation time was estimated to be 240 min.

The post-operative course of the recipient patient was successful, and she immediately became dialysis-independent with excellent renal function. Post-operative pain was well controlled by over-the-counter pain medication such as Paracetamol. She was discharged on the fifth post-transplantation day with good renal function. After the operation, the recipient did not suffer from any urinary tract infections or other relevant complications. At the ten-month follow-up, the serum creatinine level was normal (67 mmol/L).

The donor was discharged on his second post-operative day, and his recovery course was also successful with good renal function at 10 months after surgery.

## 4. Discussion

Nephrectomy of the native kidney is often required in patients who are considered candidates for a KT for several indications [1,8,9,11,12]. When NN is required, different surgical strategies should be proposed based on chronic renal diseases, the recipient’s clinical status, the residual diuresis and the type of transplantation such as deceased-related or living-related transplantation [1]. In KT candidates, the NN may be performed either as a staged approach, namely, before transplantation, or simultaneously to KT or after KT [13].

A staged NN was proposed particularly for patients with adult polycystic kidney disease (APKD) who are on the waiting list for deceased-related KT due to the unpredictability of graft availability [14]. However, the staged surgery may prevent a significant hemodynamic instability due to the NN, especially if a bilateral nephrectomy is required in APKD patients, which could negatively affect the graft reperfusion and outcomes. Additionally, a nephrectomy before transplantation makes patients non-nephric, introducing a period of dialysis and potentially sensitizing the patient due to the need for blood transfusions.

Nephrectomy after transplantation avoids dialysis and sensitization but can potentially affect the kidney allograft function and add the morbidity and cost of two separate procedures.

Indeed, a simultaneous nephrectomy at the time of the kidney transplantation avoids dialysis and two separate surgical interventions. Moreover, the use of immunosuppressors at the time of the transplantation minimizes the sensitization in case a blood transfusion is necessary. There are potential risks related to perioperative hypotension and surgical complications from the larger surgery, such as a prolonged anesthetic that could affect the allograft function and early perioperative complications. A simultaneous ipsilateral NN and KT through an open surgical approach requires a large incision and is inevitably a more debilitating operation [11]. To avoid these complications, the mini-invasive approach for NN was proposed in recent decades, and was associated with less post-operative morbidity [15]. Recently, Abrol et al. demonstrated that the laparoscopic approach for simultaneous bilateral NN with KT is safe and feasible for a living donor KT [16]. Interestingly, compared to patients receiving only KT, those who underwent simultaneous bilateral NN and KT showed a longer cold ischemia time, required more ICU care, had an increased amount of blood transfusions and had longer hospital stays. However, the authors showed that the kidney function was similar in the first year in both groups, with no difference in the delayed graft function, re-admissions or severe grade III and IV complications in the three months after surgery. Thus, all the reports on the minimally invasive approach for NN simultaneous to KT were performed using different surgical incisions and significantly prolong operating times.

To the best of our knowledge, the current case report is the first case of simultaneous mini-invasive NN and KT using the same incision. The decision of performing a simultaneous NN and LDKT was indicated by the potential benefit of limiting the exposure of our patient to the risks of prolonged time on dialysis and the related risk of suspension, which is demonstrated as being associated with higher risks of complications and overall adverse events [12]. In this case, we were conscious of the relatively small window for transplantation caused by recurrent infections and previous procedures. The prophylactic antibiotic treatment was carried out for one week before the LDKT, allowing for a safe procedure afterward in the surgical state.

In this case, a risk minimization strategy led us to avoid a retroperitoneal approach on the same side of the NN. Our intention was to avoid a fibrotic retroperitoneal space. Additionally, we felt that the risk of contamination of the transplant field with an accidental spillage of infected urine was quite substantial, particularly when removing a chronically infected kidney.

The trans-peritoneal approach from a hand port placed in the RIF was facilitated by the patient being particularly slim, allowing for a comfortable reach of the left kidney. The camera and working port were placed as per normal practice when performing a laparoscopic hand-assisted donor nephrectomy (LHA-DN). The trans-peritoneal approach proved to be particularly advantageous, allowing for an uncomplicated procedure and safe closure of the peritoneum before initiating the transplant.

We believe that in selected patients, this approach might be feasible and beneficial. In our routine practice, we consider a simultaneous NN using the hand-assisted approach followed by KT through the same incision in non-obese patients who need a native nephrectomy and can perform the kidney transplant in the contralateral iliac fossa. However, this approach might face some issues that need to be considered during the surgical plan. Firstly, the NN via a laparoscopic hand-assisted technique through an incision in the iliac fossa might be challenging in an obese patient, since reaching the upper pole of the kidney might be difficult. In such cases, a full laparoscopic nephrectomy might be more feasible. Secondly, the size of the kidney to be removed also needs to be considered. A large polycystic native kidney might not be removable by the incision in the iliac fossa, and closing the peritoneum might be difficult. Finally, this technique is more reproducible when a living donor is available in contrast to a KT from a deceased donor, since this approach might prolong the operation time in an urgent setting.

Although equivalent benefits are generally accepted in evaluating trans-peritoneal and retroperitoneal approaches in LHA-DN [17], in this case of LHA-NN, a retroperitoneal approach would not have been technically possible. Our case suggests an uncommonly reported advantage of the trans-peritoneal versus the retroperitoneal approach. Undoubtedly, a full laparoscopic or robotic procedure with an extraction of the kidney from the incision in the right iliac fossa might have been feasible, provided that the duration of the NN would not excessively impact the subsequent LDKT. As reported by other series [18], the use of the same incision also had a benefit in terms of post-operative recovery and aesthetic results, which are beneficial factors in a young woman. Of course, the current report is only a single experience, and further data are needed to confirm the benefits of this surgical approach.

This case highlights that it might be possible to perform procedures simultaneously with KT, combining safety and the least debilitating strategy, in the context of living donation as well, which allows for a detailed planning of the surgical strategy. We believe that having the availability of a comprehensive portfolio of surgical techniques represents a major benefit, allowing for more flexibility in the surgical strategy, and ultimately benefitting the final patient outcome [19].

## Figures and Tables

**Figure 1 reports-06-00029-f001:**
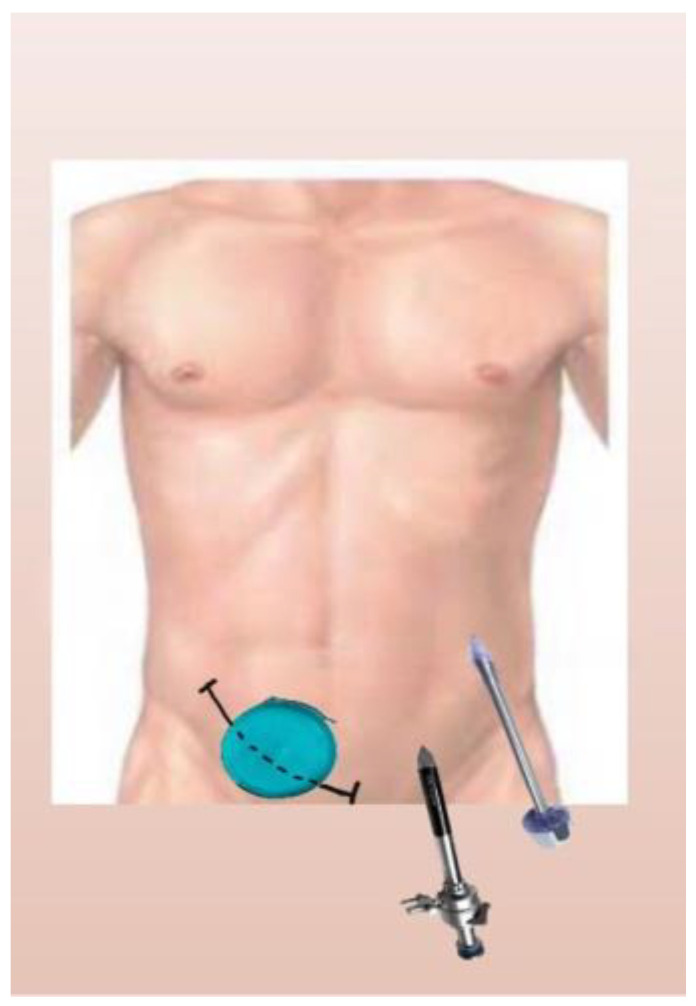
Diagram of hand port and trocars’ position.

**Figure 2 reports-06-00029-f002:**
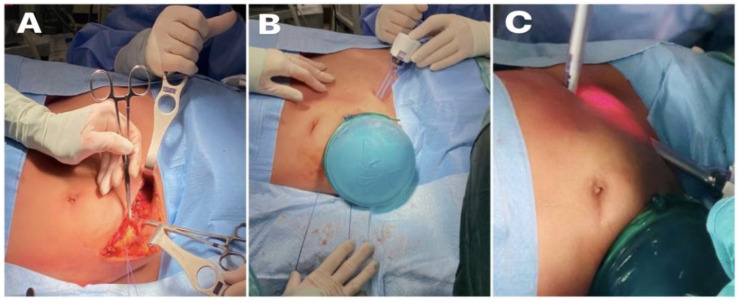
(**A**) Curved right iliac fossa incision; (**B**) hand port in right iliac fossa; (**C**) two 10 mm trocars positioned in the left iliac fossa and left upper quadrant, respectively.

**Figure 3 reports-06-00029-f003:**
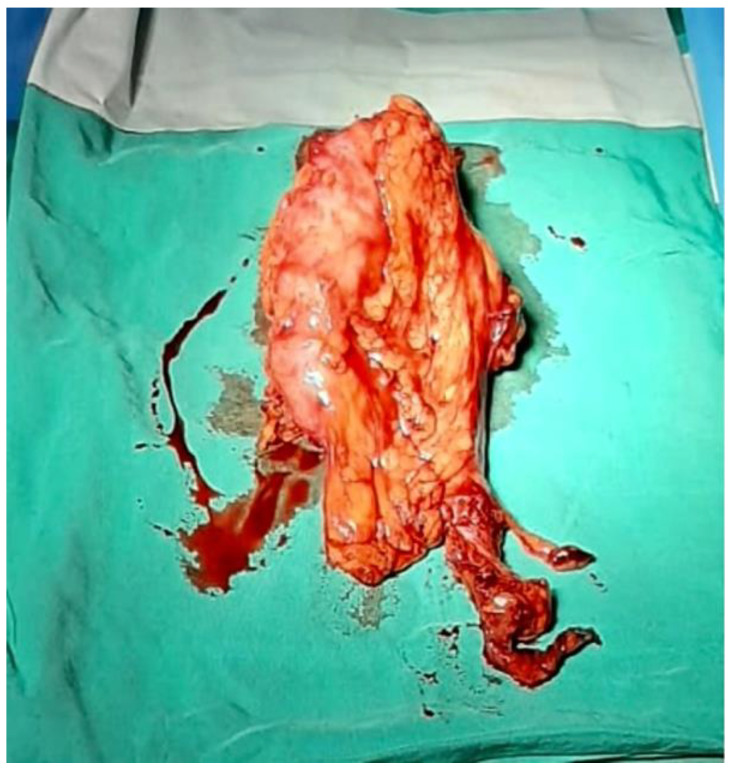
The left kidney extracted through the hand port in right iliac fossa.

**Figure 4 reports-06-00029-f004:**
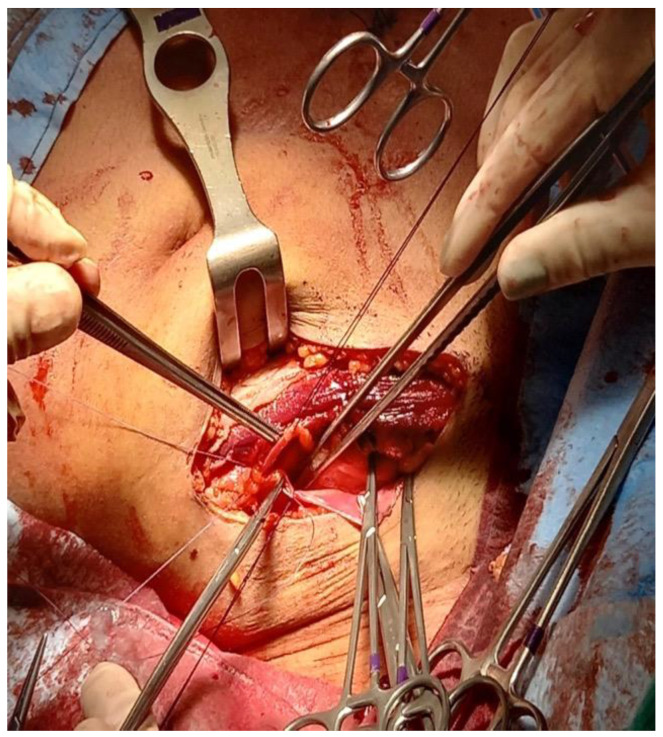
Closure of peritoneum to allow a retroperitoneal approach to the iliac vessels.

**Figure 5 reports-06-00029-f005:**
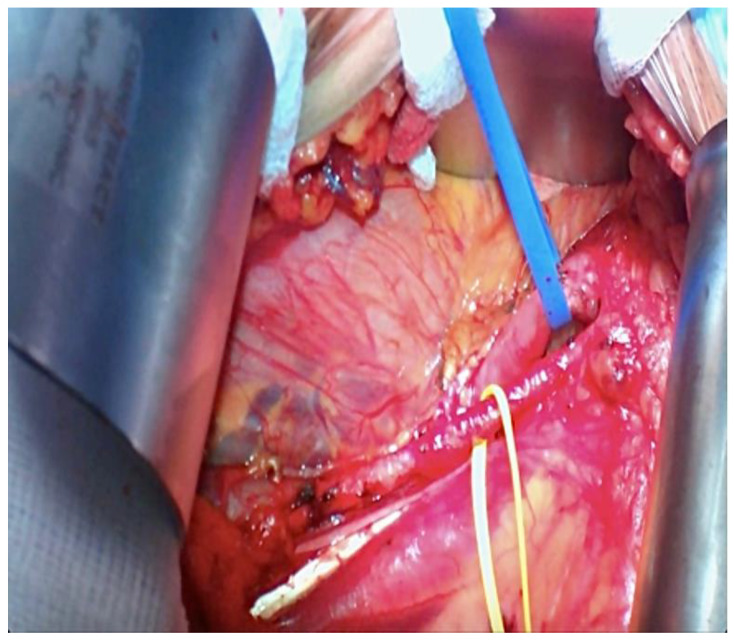
Iliac Vessels prepared for kidney transplant (the yellow loop is around the external iliac artery and the blue loop is on the external iliac vein).

**Figure 6 reports-06-00029-f006:**
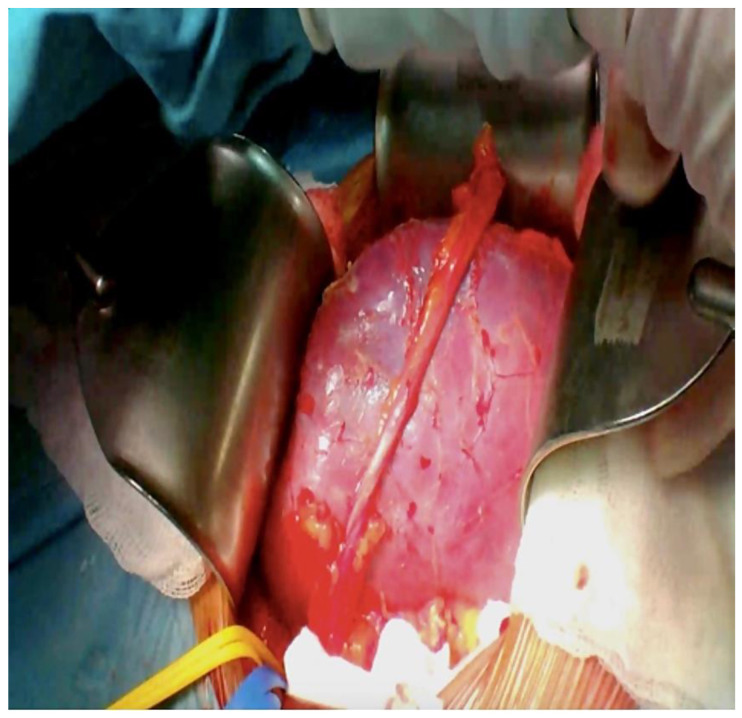
Kidney transplant appearance after re-perfusion with blood.

## Data Availability

The data presented in this study are available on request from the corresponding author. The data are not publicly available due to privacy.

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
