# Peer review of "Same Incision for Simultaneous Laparoscopic Hand-Assisted Native Nephrectomy Contralateral to the Site of the Kidney Transplant"

_reports, 2023, doi:10.3390/reports6020029_

Round 1
Reviewer 1 Report
English needs to be improved
Need a better description of the surgical approach
Need to point out the cons of this approach and reflection on its potential use in more patients (is it generalizable to other cases?)
Elaborate on the future steps of this project. Is it planned to do a study with more patients and report on the follow up?
English needs to be improved.
There are several errors
Author Response
REVIEWER 1
- English needs to be improved
Author’s reply: Thank you for your suggestion. An English language editing by an official English service was made to the revisited version.
- Need a better description of the surgical approach
Author’s reply:
A further description of the surgical approach was provided in the revisited version.
- Need to point out the cons of this approach and reflection on its potential use in more patients (is it generalizable to other cases?)
Author’s reply:
Thank you for your comment. We believe that in selected patients this approach might be feasible and beneficial. In our routine practice we consider a simultaneous native nephrectomy by hand-assisted approach followed by kidney transplantation by the same incision in non-obese patients, who need a native nephrectomy and can perform the kidney transplant in the contralateral iliac fossa. However, this approach might face some issues that need to be considered at the surgical plan. Firstly, the native nephrectomy by a laparoscopic hand-assisted technique trough an incision in the iliac fossa might be challenging in obese patient, since reaching the upper pole of the kidney might be difficult. In such cases a full laparoscopic nephrectomy might more feasible. Secondary, the size of the kidney to be removed need also to be considered. Large polycystic native kidney might not be removable by the incision in the iliac fossa and closing the peritoneum might be difficult. Finally, this technique is more reproducible when a living-donor is available as in a contest of a kidney transplantation from deceased donor since this approach might prolong the operation time in an urgent contest. All these comments have been included in the revisited paper.
- Elaborate on the future steps of this project. Is it planned to do a study with more patients and report on the follow up?
Author’s reply:
As this first case was very successful, we now consider a simultaneous native nephrectomy by hand-assisted approach followed by kidney transplantation by the same incision in all non-obese patients, who need a native nephrectomy and can perform the kidney transplant in the contralateral iliac fossa. We are getting more experience by this approach to report a case-series and longer-follow up in the next future. We are also planning to compare the results of this technique to these obtained by the sequential approach. This comment has been added in the revisited manuscript.
Reviewer 2 Report
Interesting case report of a simultaneous hand assisted lap contralateral native nephrectomy and living unrelated donor kidney transplant. The English is actually pretty good but not quite good enough. The authors need to identify a good English writer to go over the paper line by line to bring the English up to standard.
Interesting case report of a simultaneous hand assisted lap contralateral native nephrectomy and living unrelated donor kidney transplant. The English is actually pretty good but not quite good enough. The authors need to identify a good English writer to go over the paper line by line to bring the English up to standard.
Author Response
REVIEWER 2
- Interesting case report of a simultaneous hand assisted lap contralateral native nephrectomy and living unrelated donor kidney transplant. The English is actually pretty good but not quite good enough. The authors need to identify a good English writer to go over the paper line by line to bring the English up to standard.
Author’s reply:
Thank you for your comments. An English language editing by an official English service was made to the revisited version.